# Modeling Bark Thickness and Bark Biomass on Stems of Four Broadleaved Tree Species

**DOI:** 10.3390/plants11091148

**Published:** 2022-04-24

**Authors:** Bohdan Konôpka, Jozef Pajtík, Vladimír Šebeň, Katarína Merganičová

**Affiliations:** 1National Forest Centre, Forest Research Institute Zvolen, 960 01 Zvolen, Slovakia; bohdan.konopka@nlcsk.org (B.K.); jozef.pajtik@nlcsk.org (J.P.); 2Faculty of Forestry and Wood Sciences, Czech University of Life Sciences Prague, 165 000 Prague, Czech Republic; katarina.merganicova@forim.sk; 3Department of Biodiversity of Ecosystems and Landscape, Institute of Landscape Ecology, Slovak Academy of Sciences, 949 01 Nitra, Slovakia

**Keywords:** bark quantity, vertical stem profile, broadleaved species, young trees, specific surface mass

## Abstract

Considering the surface of individual tree compartments, it is obvious that the main portion of bark, i.e., the largest area and the greatest bulk mass, is located on the stem. We focused on basic bark properties, specifically thickness, surface area, biomass, and specific surface mass (expressed as dry weight per square unit) on stems of four broadleaved species: common aspen (*Populus tremula* L.), goat willow (*Salix caprea* L.), rowan (*Sorbus aucuparia* L.), and sycamore (*Acer pseudoplatanus* L.). Based on the previous work from mature forests, we hypothesize that bark properties of young trees are also species-specific and change along the stem profile. Thus, across the regions of Slovakia, we selected 27 forest stands composed of one of the target broadleaved species with ages up to 12 years. From the selected forests, 600 sample trees were felled and stem bark properties were determined by measuring bark thickness, weighing bark mass after its separation from the stem, and drying to achieve a constant weight. Since the bark originated from trees of varying stem diameters and from different places along the stem (sections from the stem base 0–50, 51–100, 101–150, 151–200, and 201–250 cm), we could create regression models of stem characteristics based on the two mentioned variables. Our results confirmed that bark thickness, thus also specific surface mass, increased with stem diameter and decreased with distance from the stem base. While common aspen had the thickest stem bark (4.5 mm on the stem base of the largest trees) the thinnest bark from the analyzed species was found for sycamore (nearly three times thinner than the bark of aspen). Since all four tree species are very attractive to large wild herbivores as forage, besides other uses, we might consider our bark mass models also in terms of estimating forage potential and quantity of bark mass consumed by the herbivory.

## 1. Introduction

Plants, which also include trees growing in forest ecosystems, are composed of a structural complex of organs characterized by their typical forms but mainly by fulfilling specific functions [1]. It is generally known that the basic compartments of trees are: assimilatory organs, branches, a stem, and roots. While assimilatory organs are covered by epidermis, woody parts, i.e., branches, a stem, and roots are covered by bark [2]. Bark is structurally much more complex than wood [2]. Considering the surface of the individual tree compartments, it is obvious that the main portion of bark, i.e., the largest area and most of its mass, is located on the stem. Previous estimates [3] of mature forest trees showed that tree bark on stems and branches represents 9 to 15% of the stem volume. Basically, tree bark consists of two components: phloem (inner part) and periderm (outer part, usually formed by cambium and cork layer; [1]). While phloem plays an essential role in translocation of carbohydrates [4], the periderm reduces water loss [5] and protects woody parts from mechanical injuries [6], climatic extremes [7], forest fires [8], fungal pathogens [9], and boring insects [10]. Apart from the protection function, tree bark serves as structural support, as nutrient and water storage, and for transport of photosynthates [11,12].

Several authors (e.g., [13,14,15,16,17]) studied bark thickness. The authors prevailingly modeled tree bark thickness at a certain location (usually 130 cm above the ground level) or along the stem profile, i.e., from the stem base to the tree top with a step of 1.0 or 2.0 m, usually using diameter at breast height (*DBH*, i.e., measured 130 cm from the ground level) as a predictor. For instance, Stängle and Dopremann [17] stated that for practical applications, diameter outside bark and *DBH* are suggested as explanatory variables of models predicting bark thickness. The authors pointed out that the interest in accurate bark thickness estimates is driven by the need to obtain not only accurate inside-bark diameters and volumes, but also precise estimates of bark volume. The importance of these estimates increased with the shift in the commercial relevance of bark from a useless residue to a valuable fuel and feedstock for high-value biomaterial [18], such as bark tannin-based foams (e.g., [19]), cork production [20], or for medicinal purposes [13]. In general, bark is richer in nutrients than wood [21]. Moreover, bark quantity estimates are also necessary for complete tree biomass estimates, which have recently become important, especially for quantifying forest carbon stocks [22].

As most of previous bark-focusing studies and models were motivated by certain economic interests (bark utilization), nearly all authors focused on mature trees. One of a few studies about bark quantity of young trees was published by Pajtík et al. [23]. In their tree-biomass specialized monograph, the authors quantified bark biomass and its contribution to the total tree biomass for eight tree species growing in the Western Carpathians region. The derived models were based on tree height and/or diameter at stem base (*D*_0_), and showed a decreasing contribution of bark mass to total stem mass as well as to total tree biomass with tree size. Here, we utilized the database which had been used for the preparation of the monograph and constructed more detailed models of bark thickness and biomass respecting not only tree size, but also the position along the vertical stem profile.

Tree bark is also one tree component that is intensively damaged by wild large herbivores (e.g., [24,25,26,27,28,29]). Broadleaved species such as common aspen (*Populus tremula* L.), goat willow (*Salix caprea* L.), rowan (*Sorbus aucuparia* L.), and sycamore (*Acer pseudoplatanus* L.) have been particularly attractive to large wild herbivores, especially red deer (*Cervus elaphus* L.). Bark browsing (also termed as bark stripping or bark peeling) and the overall impact of large wild herbivory on forest vegetation has been increasing in many European countries because of continuously rising population density of these animals, especially red deer (e.g., [29,30,31]). Therefore, we need better knowledge about bark biomass on stems, which can be regarded as a forage source available on tree vegetation for wild game. In fact, many previous studies focused on stem bark browsing quantification based on simple indicators such as browsing rate at a stand level or bitten off area at a single tree level [27,30,32]. However, so far, there is not any precise quantification on browsed bark biomass either at a tree or at a stand level. The main reason for this is the lack of bark models usable for the estimation of browsed stem bark biomass, or possibly data concerning stem bark biomass available for browsing (forage potential).

This paper aims to model stem bark thickness and bark mass of four broadleaved species along the vertical stem profile. Based on the previous work from mature forests, we hypothesize that despite small tree age, bark properties of young trees are species-specific and change along the stem profile. However, we assume that interspecific differences in the basic bark traits do not coincide with those already known from studies of mature trees. We derive regression models to estimate bark thickness and bark mass on the basis of diameter at stem base that can be used in further biomass studies. Subsequently, we construct a regression model to determine the specific surface mass that expresses bark amount per square unit based on stem diameter. This model would be applicable, for instance, in the estimates on stem bark browsing by large wild herbivores.

## 2. Results

First, we derived relationships between diameters *DBH* and *D*_0_ (Figure 1 and Table 1) and between tree height *H* and diameter *D*_0_ (Figure 2 and Table 2). The results indicate rather large differences in the relationships between species. The first relationship shows that the order of trees regarding tapering in the lower part of stem (from a cylindrical to a conic shape, also Table 3) is: goat willow, sycamore, rowan, and common aspen.

As for the second relationship, we can see that the descending order of species with respect to their slenderness ratios (i.e., the ratio between tree height and stem diameter) is: aspen, sycamore, rowan, and goat willow (Figure 2). These relationships provide us not only with the information about stem shapes but also create a possibility for potential users to use *DBH* or tree height instead of diameter *D*_0_ for further estimates on bark characteristics using the regression characteristics in Table 2.

In the next step, we derived species-specific regressions for using Equation (5) to calculate stem radii at different heights on stems. The models were highly significant and explained from 59 to 91.6% of radius variation along the stem (Table 3).

Next, we analyzed bark thickness along the stem profile. Bark thickness varied between approximately 0.5 to 4.5 mm depending on the species and height on stem (Figure 3 and Table 4). There were evident differences in bark thickness between tree species and between individual stem heights (distances from the ground level to 50, 100, 150, 200, and 250 cm). The thickest bark was found for common aspen, followed by substantially thinner bark for goat willow, rowan, and sycamore. The results suggest that while the differences between the bark thickness at different heights on the stem of common aspen and goat willow were large, especially when comparing the bottom 50 cm with the others, the values of bark thickness of rowan and sycamore along their vertical stem profiles were similar.

Further analyses proved large differences in the surface area of bark between tree species (Figure 4). Note that the bark surface area was derived from the modeled values of radii at specified heights above the ground using Equation (5) (models shown in Table 3) and the formula for calculating the surface of a truncated cone (Equation (4)). The descending order of tree species regarding the surface area of bark follows: common aspen, sycamore, rowan, and goat willow. Additionally, in this case, we can see large differences between the lowest stem section and the other sections.

Bark mass was obtained from the information on bark thickness and its surface area using Equation (6) (Figure 5). The largest values were found for common aspen, for which the maximum value was nearly 350 g (for a diameter *D*_0_ of 100 mm and 0–50 cm section from the ground). For all tree species, large differences in bark mass were recorded between the lowest section 0–50 cm and all the other sections.

Finally, specific surface mass was calculated respecting both *D*_0_ and individual stem sections (Figure 6). The largest values were found for common aspen and the lowest values for sycamore. For instance, while common aspen with *D*_0_ of 100 mm had the specific surface mass in the lowest stem part, about 22 g per dm^2^, sycamore with the same diameter had a specific surface bark mass of only about 7 g per dm^2^ in the same stem section. Interestingly, differences in specific surface mass of bark between the lower stem part (0–50 cm from the ground level) and upper stem parts exhibited much more contrast for common aspen and goat willow than for rowan and sycamore.

## 3. Discussion

Our results clearly demonstrate that from the intraspecific point of view, bark thickness varies with stem diameter as well as along the vertical stem profile. In fact, both of these variables are prevailingly related to bark age, i.e., older bark (on older trees and/or bark situated at a lower stem part) is thicker than younger bark (younger trees and/or at higher stem parts). In general, each tree stem contains a layer of dividing cells called the vascular cambium [2]. These cells ultimately differentiate into either bark or wood tissues, and over time they are responsible for the radial growth of the stem [33].

Moreover, large differences in bark thickness occur between the species. The differences between the species with the thinnest bark (i.e., sycamore) and the thickest bark (common aspen) are nearly three-fold. Trees are sometimes categorized as thin-barked and thick-barked (e.g., [34]), although this categorization might be irrelevant in the case of old trees, when most species have already massive, cracked or furrowed bark. On the other hand, our results coincide with the work of Zhao et al. [35], which focused on interspecific differences in bark thickness of adult trees. The authors show values of 0.7, 0.4, and 0.3 cm in aspen, goat willow and rowan, respectively. It means that the order of the species from the point of bark thickness is the same as that shown by our models.

Bark thickness of all investigated species decreased with height on stems (distance from ground), which is consistent with studies on other species [12,36]. The lowest variability in bark thickness along the stem profile (specifically the bark in 0–50 cm section versus the bark in 200–250 cm from the ground level) was found for sycamore and the highest for common aspen. The large variability in the case of common aspen might be caused by the process of early “roughening up” of the bark on the stem base. Hence, the species with thick bark have also large variability in bark thickness along the vertical stem profile. This is logical in terms of large annual bark increment and consequently large differences between bark parts of different ages, hence also along the vertical profile.

We could not find any results about bark thickness for the four studied broadleaved tree species in other works, except for common aspen [37]. The paper showed that bark thickness (measured at a height of 130 cm from the ground level) of aspen increased with *DBH*, while it measured approximately 2.5 mm for trees with *DBH* of 10 cm, and doubled in trees with *DBH* of 20 cm. Larger trees with *DBH* of 30 cm and more had similar values of bark thickness equal to nearly 9 mm. If we consider bark thickness (2.5 mm) for the lower limit of the stem diameter (10 cm) shown in the paper by Johansson [38], our results manifested very close absolute values, specifically 3.0 mm. Actually, while some authors modeled tree thickness using exclusively stem diameter (e.g., [13,16,37,38,39]), a few implemented tree age (e.g., [40]) or a combination of stem diameter and tree age (e.g., [41]). The findings clearly suggest that trees with the same diameter but of different ages may contain differently thick bark, specifically older trees having thicker bark than younger ones [41]. Another factor considered when studying bark thickness was the bio-sociological position of trees in a stand [42]. The authors found that the bark of co-dominant, intermediate, and suppressed trees of *Sequoia sempervirens* (D. Don) Endl. was approximately 8, 14, and 18% thicker than the bark of dominant trees with the same dimensions, respectively. However, the authors did not show exact ages of the individual trees, which were compared and the differences might be related primarily to different tree ages. At the same time, Berill et al. [42] proved that bark thickness is related to tree genotype, stand structure, and latitude, which indicates that intraspecific variability of bark thickness is influenced by a variety of internal and external factors; hence, further research should be conducted in this field.

A positive relationship between tree size and stem bark thickness has been previously interpreted to mean that bark thickness is strongly influenced by its defensive role [14]. According to this assumption, which considers that bark production is costly, defensive properties (especially insulation and herbivory-protection) of bark should be secured by a certain value of its thickness, beyond which additional thickness brings no additional advantages [43]. This knowledge points out at bark thickness as a type of cost–benefit optimization resulting from the evolution at species level. Regardless, as for the relationship between bark thickness and potential browsing by large wild herbivory, there are no clear scientific results so far, particularly regarding the maximum bark thickness which might be bitten off by certain herbivory species.

The findings proved that common aspen had between two and four times higher specific surface mass of bark than other investigated broadleaved species. Results on specific surface mass of bark are missing in literature, except for a paper by Konôpka et al. [44] that studied young trees of common ash (*Fraxinus excelsior* L.). The authors showed that the specific surface mass of bark for ash was between 4.0 and 7.5 g per dm^2^ for trees with *D*_0_ between 20 and 60 mm [44]. Here, we consider that while sycamore and common ash are commercially usable tree species, common aspen, goat willow, and rowan are usually not of industrial interest. Therefore, these so called “pioneer” species, mainly common aspen (due to its high values of specific surface mass), can create a forage base for red deer and other large wild herbivory, attracting them from some commercially important tree species such as *Fagus sylvatica* and *Picea abies* (see also [45]). On the other hand, we assume that common aspen (and possibly also goat willow in the lowest parts of stem) can have very thick and furrowed bark already from young growth stages, which might prevent herbivory browsing. So far, however, this has not been experimentally proven. Hence, the relationship between herbivory bark browsing and tree species of different sizes with regard to a variety of aspects should be further studied.

Our research can be understood as both a pilot and a tentative one. It can be regarded as a pilot study because previous work that applies modeling of bark properties focused almost exclusively on large trees with a minimum stem diameter of about 10 cm (e.g., [16,42,46]). We decided to perform measurements and model bark properties for small trees. Moreover, we derived new characteristics of bark properties, i.e., specific surface mass. This indicator might be possibly utilized further for researching issues of bark browsing by large wild herbivores. This kind of study would be conducted in terms of estimates on forage potential, but also quantification of actually consumed bark biomass by herbivores at a tree or a stand level. The reason why we labeled the study as tentative is in its incompleteness, specifically not covering the full range of tree sizes, i.e., from very initial to mature individuals. Stem growth rate (of both wood and bark) can also be an important factor influencing bark properties. Hence, much more work must be done in the field of modeling bark characteristics, including tree age as a predictor.

## 4. Materials and Methods

### 4.1. Sampling and Data Collection

Our sampling covered four broadleaved tree species, namely: common aspen, goat willow, rowan, and sycamore. Firstly, we selected naturally regenerated even-aged forest stands for each target tree species with a maximum age of 12 years. These were from the current national database of forest stands and based on data from Forest Management Plans (see also http://gis.nlcsk.org/lgis/, accessed on 10 February 2022). From these stands, we further chose forest stands that grew at moderately fertile sites; therefore, nearly all of them were found on mesotrophic cambisols. Since mesotrophic cambisols [47] occur within 70% of forest area in Slovakia, we focused our activities exclusively on these soil conditions.

After the tentative selection of forest stands, we personally inspected the individual stands and selected between five and ten stands per species based on the following criteria: contribution of target species to stand density of at least 90%; stand compactness, i.e., no serious treeless gaps; none or negligible damage by any pests; and self-development since forest establishment, i.e., no past silvicultural measures (e.g., tending) during the existence of the selected stand.

Field work was conducted each year between 2016 and 2020, always in the second half of the growing season. Each tree species at a specific location was represented by 15–25 individuals. Thus, totally, we sampled between 100 and 200 trees per species (Table 5). Trees were taken from a variety of territories of Slovakia, specifically common aspen from the orographic units Kremnické vrchy, Štiavnické vrchy, Krupinská planina, Malá Fatra, and Nízke Tatry. The samples of goat willows originated from Kysucké Beskydy and Vysoké Taty, the samples of rowan from Vysoké Tatry and Podtatranská kotlina, and the samples of sycamore were from Malé Karpaty, Strážovské vrchy, Kysucké Beskydy, Javorie, Poľana, Nízke Tatry, and Slovenské rudohorie.

Sampled trees were randomly selected in each stand while avoiding damaged, dying, deformed, or atypically shaped individuals, e.g., ones growing at forest stand edges. We tried to choose individuals of all bio-sociological positions, i.e., dominant, co-dominant, intermediate, and suppressed ones. However, we omitted severely suppressed individuals with the symptoms of initial die-back.

Each selected tree was felled at the ground level. Diameters at stem base (*D*_0_) and at breast height (*DBH*) were measured using a digital caliper with ±0.1 mm precision. Then, all branches were cut off by garden scissors and disposed. A precise tree height (±1.0 cm) was measured using a metal tape measure. Afterward, all tree stems were packed in labeled paper bags and transported to our laboratory (see also Konôpka et al. [48]).

The stems were divided into several sections, while the number of sections depended on the tree height (more sections were taken from higher trees). When dividing the tree into sections, we implemented two principles: (1) to have at least 4 sections per tree, and (2) not to exceed the length of 1 m, which was set as the maximum length of a section. Diameters of each section were measured with a digital caliper (precision ±0.1 mm) in two perpendicular directions at its both ends, and in the middle. The measurements were repeated twice: before peeling the bark off (i.e., over bark) and after peeling the bark (under bark). Bark thickness at a certain position was calculated as a difference between the diameters before and after bark peeling.

At a tree level, we took overall bark (see also Pajtík et al. [23]) that were dried in a large-capacity drying oven at a temperature of 95 *°*C for 120 h to reach the constant weight. Afterward, they were weighed with an accuracy of 0.1 g.

### 4.2. Data Processing and Modeling

The relationship between tree height and stem base diameter was described with the following regression:(1)$$H=\frac{D_{0^{2}}}{b_{0}+b_{1}D_{0\;}+b_{1}D_{0\;}}$$
where:*H* is a tree height (m);*D*_0_ is a diameter at stem base (mm);*b*_0_, *b*_1_, *b*_2_ are regression coefficients.

The relationship between diameter at breast height and stem base diameter was described with the following linear function:*DBH = b*_0_ + *b*_1_*D*_0_
(2)

where:*DBH* is a diameter at breast height (mm);*D*_0_ is a diameter at stem base (mm);*b*_0_, *b*_1_ are regression coefficients.

Values of regression coefficients and statistical characteristics of both relationships are presented in Table 2. Here, we would like to point out that our modeling of bark properties was based exclusively on *D*_0_ (independent variable). Thus, relationships between *D*_0_ and tree height (or diameter *DBH*) also create the possibility for potential users to implement these tree characteristics for estimating bark characteristics.

**Bark thickness** at a certain height above the ground was calculated using a power function with two predictors:*T_b_ = b*_0_ *D*_0_*^b^*^1^*H_g_ h^b^*^2^(3)
where:*T_b_* is bark thickness (mm);*D*_0_ is a diameter at stem base (mm);*H_g_* is a distance from the ground level (cm);*b*_0_*, b*_1_, *b*_2_ are regression coefficients.

The model for the calculation of bark thickness based on the height above the ground and the stem base diameter *d*_0_ was derived from the measured data on bark thickness at specific heights. The heights were not constant for the whole dataset, because the sections were of different lengths. The model of bark thickness was derived only for the bottom part of the stem up to the height of 2.5 m.

**Bark surface** was calculated for individual stem sections using the formula for calculating the surface of a truncated cone, omitting the areas of base and top circles:(4)$$S_{b}=\pi \left(r_{1}+r_{2}\right)\sqrt{{\left(r_{1}-r_{2}\right)}^{2}+l_{s}{}^{2}}$$
where:*S_b_* is bark surface (cm^2^);*r*_1_ is a radius of the bottom end (cm);*r*_2_ is a radius of the top end (cm);*l_s_* is the length of the section (cm).

To determine the radii *r*_1_ and *r*_2_ at any height above the ground (*H_g_*), we derived a continuous regression model for stem radius with *D*_0_ and *H_g_* as predictors. This relationship was also derived for a stem part up to 2.5 m above the ground. The formula is:(5)$$r=b_{0}D_{0}^{b_{1}}Hg^{b_{2}}$$
where:*r* is a stem radius (mm);*D*_0_ is a diameter at stem base (mm);*Hg* is a distance from the ground level (cm);*b*_0_, *b*_1_, *b*_2_ are regression coefficients.

Model (5) allowed us to determine *r*_1_ and *r*_2_ for a tree with a diameter *D*_0_ at predefined heights above the ground (0, 50, 100, 150, 200, and 250 cm). The calculated values of *r*_1_ and *r*_2_ were then used in Equation (4) to derive the bark surface of sections 0–50, 51–100, 101–150, 151–200, 201–250 cm. Models (3) and (5) were derived in forms of allometric equations in very similar ways as previously published by Marklund [49], Ledermann and Neumann [50], and Cienciala et al. [51].

**Bark mass** of individual sections was derived using the equation:*W_b_ = V_b_·ρ_b_*(6)
where:*W_b_* is bark mass weight (g);*V_b_* is bark volume (cm^3^);*ρ_b_* is bark density (kg per m^3^).
where bark volume *V_b_* of each section was calculated as a difference between the section volume outside bark and the volume inside bark. The section volume (cm^3^) was determined using the volume equation for a truncated cone:*V_b_ = 1/3πl_s_ (r_1_*^2^ + *r*_1_*r*_2_ + *r_2_*^2^*)*
(7)

where the symbols are the same as in Equation (4).

The species-specific values of bark density *ρ* were taken from the only available paper by Pajtík et al. [23]. Bark mass was calculated under the assumption that bark density is constant along the whole height profile 0–250 cm (see also [23]).

**Specific surface mass** of bark was calculated as follows:*SPH = W_b_/S_b_*(8)
where:*SPH* is specific surface mass of bark (g per dm^2^);*W_b_* is bark mass (g);*S_b_* is bark surface (dm^2^).

## 5. Conclusions

This study evaluated vertical changes in bark properties along the stems of four broadleaved tree species in the Western Carpathian Mountains., Slovakia. The results confirmed that bark thickness is species-specific and changes with tree size and height on stem. The thickest bark was found for common aspen. Goat willow, rowan, and sycamore had much thinner bark than aspen (by approximately 2, 2.5, and 3 times, respectively). Our results revealed the same order of tree species according to bark thickness as in the case of mature trees published in other previous works. The bark thickness was greatest at the bottom part of trees and decreased with the increasing height on stems, irrespective of species. Hence, biomass studies of young trees should consider the changes along the vertical stem profile. The models derived in this study account for these changes and may be used under similar growing conditions in Central Europe.

The revealed substantial differences between bark properties of tree species indicate that using generic values of bark thickness or bark mass may cause significant bias in estimated biomass. Therefore, species-specific and, if possible, also local models that account for tree size should be implemented for tree biomass estimates including stem bark. The new information about bark properties can be considered as another scientific contribution of our work because a substantial part of previous studies did not consider stem bark as a separate tree component but often included it in over-bark stem estimates. Finally, we would like to conclude that findings on the thickness and/or biomass of stem bark should be not considered only as an output for estimating potential residues in the wood processing industry. Such novel knowledge could be applied in studies of stem bark utilization for humankind and as a part of large wild herbivory nutrition.

## Figures and Tables

**Figure 1 plants-11-01148-f001:**
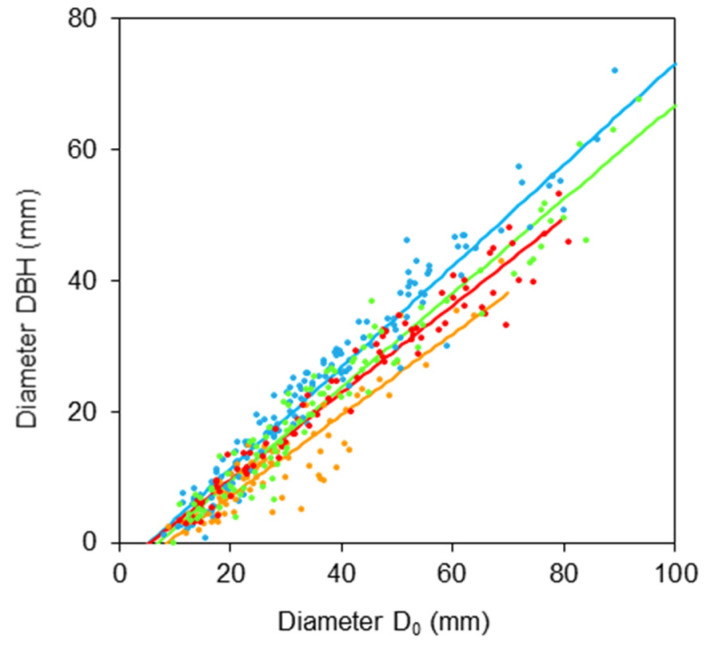
Relationship between diameter at breast height *DBH* and stem base diameter *D*_0_ for common aspen (blue), goat willow (orange), rowan (red) and sycamore (green), see Table 1 for formula and statistical characteristics.

**Figure 2 plants-11-01148-f002:**
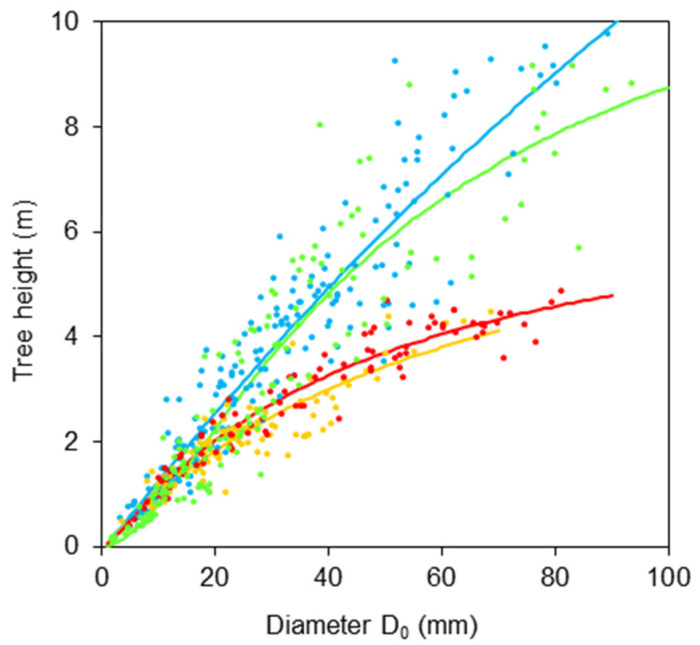
Relationships between tree height and stem base diameter *D*_0_ for common aspen (blue), goat willow (orange), rowan (red) and sycamore (green), see Table 2 for formula and statistical characteristics.

**Figure 3 plants-11-01148-f003:**
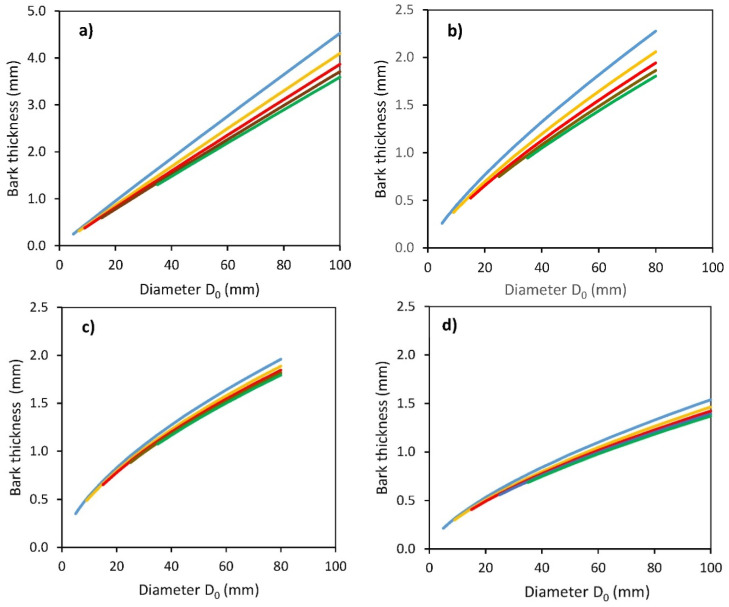
Modeled relationships between bark thickness and stem base diameter *D*_0_ for common aspen (**a**), goat willow (**b**), rowan (**c**) and sycamore (**d**), at different predefined vertical distances *H_g_* from the ground level (blue: *H_g_* = 50 cm, orange: *H_g_* = 100 cm, red: *H_g_* = 150 cm, brown: *H_g_* = 200 cm, green: *H_g_* = 250 cm), see Table 4 for formulas and statistical characteristics of derived regression models.

**Figure 4 plants-11-01148-f004:**
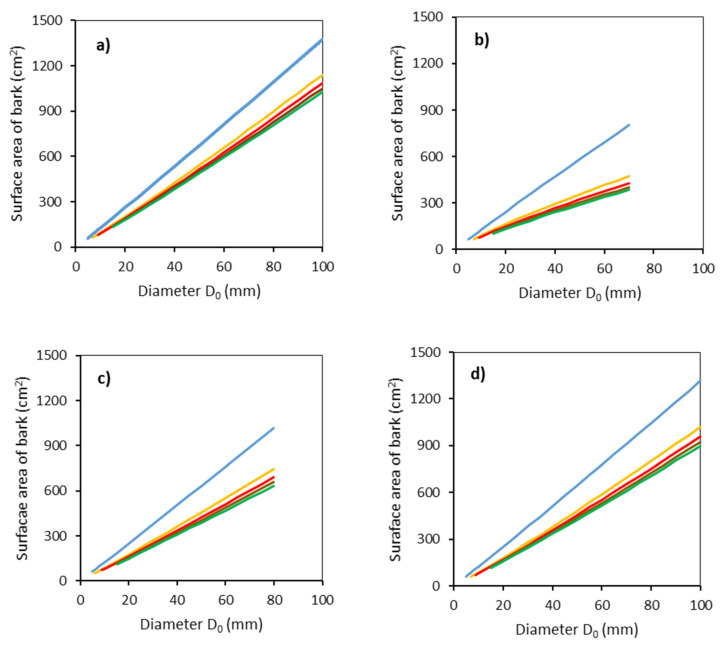
Modeled relationships between surface bark area and stem base diameter *D*_0_ for common aspen (**a**), goat willow (**b**), rowan (**c**) and sycamore (**d**), in separate stem sections defined by the vertical distance from the ground level (blue: 0–50 cm, orange: 51–100 cm, red: 101–150 cm, brown: 151–200 cm, green: 201–250 cm) using Equation (4) and modeled stem radii (Equation (5) and Table 3).

**Figure 5 plants-11-01148-f005:**
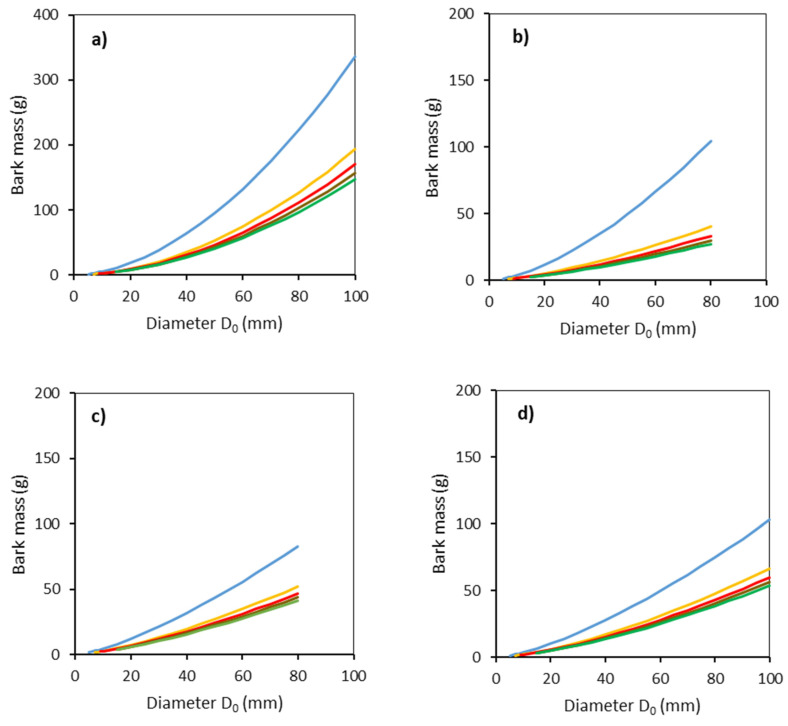
Modeled relationships between bark mass and stem base diameter *D*_0_ for common aspen (**a**), goat willow (**b**), rowan (**c**) and sycamore (**d**), in separate stem sections defined by the distance from the ground level (blue: 0–50 cm, orange: 51–100 cm, red: 101–150 cm, brown: 151–200 cm, green: 201–250 cm) using Equation (6) and modeled stem radii (Equation (5) and Table 3).

**Figure 6 plants-11-01148-f006:**
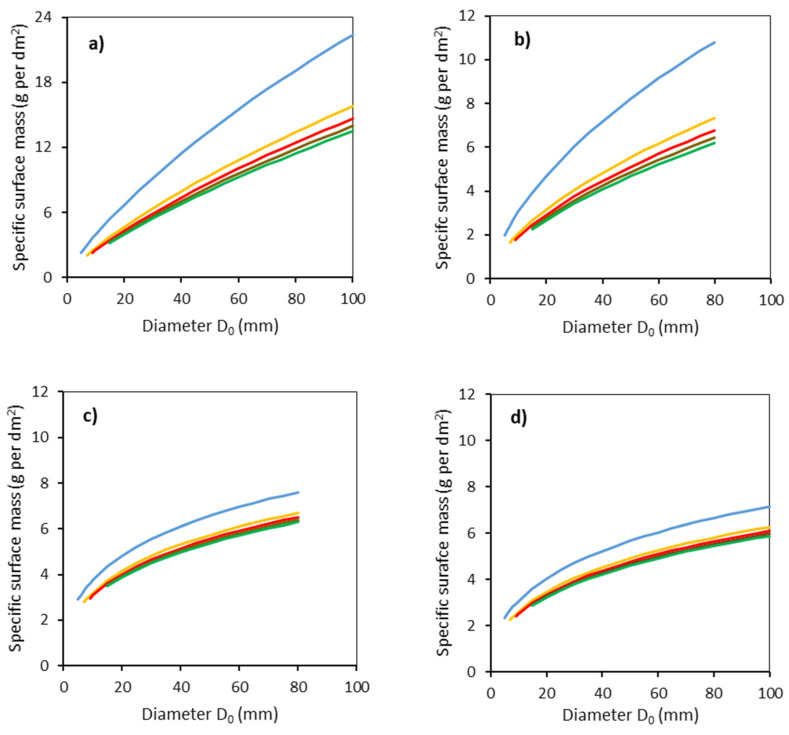
Modeled relationships between specific surface bark mass and stem base diameter *D*_0_ for common aspen (**a**), goat willow (**b**), rowan (**c**) and sycamore (**d**), in individual stem sections defined by the vertical distances from the ground level (blue: 0–50 cm, orange: 51–100 cm, red: 101–150 cm, brown: 151–200 cm, green: 201–250 cm) using Equation (8) and modeled stem radii (Equation (5) and Table 3).

**Table 1 plants-11-01148-t001:** Relationships between diameter at breast height *DBH* and stem base diameter *D*_0_ for four broadleaved tree species explained by the formula: *DBH = b*_0_ *+ b*_1_*D*_0_ where *b*_0_, *b*_1_ are regression coefficients, *S.E*. their standard errors, coefficient of determination (*R*^2^), mean squared error (*MSE*). All calculated *p*-values of derived regression coefficients were <0.001. See Figure 1 for model visualization.

Tree Species	*b* _0_	*S.E.*	*b* _1_	*S.E.*	*R* ^2^	*MSE*
Common aspen	−3.422	0.911	0.763	0.022	0.890	29.46
Goat willow	−5.285	0.971	0.620	0.031	0.838	11.60
Rowan	−3.629	0.711	0.665	0.016	0.958	7.83
Sycamore	−4.948	0.732	0.718	0.017	0.950	12.38

**Table 2 plants-11-01148-t002:** Relationship between tree height H and stem base diameter *D*_0_ for four broadleaved tree species described by the formula: *H = D*_0_^2^*/(b*_0_ *+ b*_1_*D*_0_ *+ b*_2_*D*_0_^2^*)*—regression coefficients *b*_0_, *b*_1_, *b*_2_, their standard errors (*S.E.*), *p*-value (*P*), coefficient of determination (*R*^2^), mean squared error (*MSE*). See also Figure 2 for model visualization.

Tree Species	*b* _0_	*S.E.*	*P*	*b* _1_	*S.E.*	*P*	*b* _2_	*S.E.*	*P*	*R* ^2^	*MSE*
Common aspen	8.221	11.812	0.487	7.077	0.693	<0.001	0.021	0.009	0.017	0.897	1218.609
Goat willow	6.921	4.691	0.143	8.127	0.542	<0.001	0.126	0.016	<0.001	0.812	237.939
Rowan	14.640	13.260	0.273	6.486	0.969	<0.001	0.135	0.014	<0.001	0.933	287.626
Sycamore	84.237	25.897	0.001	3.273	1.311	0.014	0.073	0.014	<0.001	0.882	723.968

**Table 3 plants-11-01148-t003:** Relationships of stem radius *r* at a specific height from the ground to stem base diameter *D*_0_ and the distance from the ground (*H_g_*) for four broadleaved tree species described by the formula: *r = b*_0_*D*_0_*^b^*^1^*H_g_* (where *r* is in cm, *D*_0_ in mm, *H_g_* in cm)—regression coefficients *b*_0_, *b*_1_, their standard errors (*S.E*.), *p*-value (*P*), coefficient of determination (*R*^2^), mean squared error (*MSE*).

Tree Species	*b* _0_	*S.E.*	*P*	*b* _1_	*S.E.*	*P*	*b* _2_	*S.E.*	*P*	*R* ^2^	*MSE*
Common aspen	0.038	0.002	<0.001	1.076	0.010	<0.001	−0.092	0.002	<0.001	0.916	0.057
Goat willow	0.088	0.010	<0.001	0.847	0.034	<0.001	−0.179	0.007	<0.001	0.594	0.107
Rowan	0.045	0.004	<0.001	1.038	0.020	<0.001	−0.139	0.003	<0.001	0.886	0.075
Sycamore	0.036	0.002	<0.001	1.079	0.014	<0.001	−0.110	0.004	<0.003	0.851	0.089

**Table 4 plants-11-01148-t004:** Relationships between bark thickness *T_b_* and diameter *D*_0_ and the vertical distance from the ground (*H_g_*) for four broadleaved tree species separately described by the formula: *T_b_ = b*_0_*D*_0_*^b^*^1^*H_g_^b^*^2^ (where *T_b_* is in mm, *D*_0_ in mm, *H_g_* in cm)—regression coefficients *b*_0_, *b*_1_, *b*_2_, their standard errors (*S.E.*), coefficient of determination (*R*^2^), mean squared error (MSE). All calculated *p*-values of derived regression coefficients were <0.001.

Tree Species	*b* _0_	*S.E.*	*b* _1_	*S.E.*	*b* _2_	*S.E.*	*R* ^2^	*MSE*
Common aspen	0.092	0.016	0.968	0.016	−0.144	0.004	0.786	0.287
Goat willow	0.129	0.010	0.784	0.023	−0.145	0.004	0.671	0.097
Rowan	0.160	0.011	0.620	0.018	−0.055	0.005	0.681	0.089
Sycamore	0.098	0.006	0.659	0.016	−0.072	0.005	0.587	0.059

**Table 5 plants-11-01148-t005:** Main characteristics of sampling locations and sampled trees of investigated broadleaved species.

Tree Species	Altitude Range (m a.s.l.)	Number of Stands	Mean Stand Ages	Number of Sampled Trees	Mean Tree Height (Standard Deviation) (m)	Mean Diameter *D*_0_ (Standard Deviation) (mm)
Common aspen	335–870	7	2–11	180	3.84 (2.45)	31.9 (21.1)
Goat willow	750–1030	5	2–12	120	2.04 (0.85)	25.0 (13.2)
Rowan	941–1122	5	2–12	100	2.82 (1.21)	36.7 (21.4)
Sycamore	415–970	10	2–12	200	2.85 (2.30)	25.8 (13.2)

## Data Availability

Not applicable.

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
