# Peer review of "Modeling Bark Thickness and Bark Biomass on Stems of Four Broadleaved Tree Species"

_plants, 2022, doi:10.3390/plants11091148_

Round 1
Reviewer 1 Report
Title:
1/ significant shortening mandatory
2/ condensate the main findings obtained from your modelling into a short and groundbreaking claim
Abstract:
3/ better follow the established schema of writing academic Abstract: A/ introduction (urgency and significance of the research hypothesis); B/ principles of the methods used + key results; C/ conclusions (commercial and environmental impacts)
4/ there is no reason to go into detail and present the results obtained under specific reaction conditions, rather provide a synthesis of the results obtained
5/ originality of these "discoveries" is questionable, similar findings can be deduced from existing literature, better highlight the novelty of your discovery (reveal the mechanisms behind your results)
Introduction:
6/ make sure that this chapter fully introduces any reader into to the topic, explain all the botanical terms, units, Latin and Greek letters, abbreviations and the whole context that is necessary for anyone (including experts from other disciplines) to understand the following chapters
7/ deeper review the latest findings in the field, comment on the industrial and economic point of view
8/ the research hypothesis is not clearly stated, better justify the urgency and importance of its investigation (clearly identify those who will benefit from the findings obtained)
Results:
9/ avoid data overkill, present only the most most industrially and environmentally important results
Discussion:
10/ show more self-criticism to your work (can all the methods and results be fully trusted? what are the weaknesses of the methods used? where do the main measurement inaccuracies arise? what are the limitations from a commercial point of view? are the lessons learned transferable to other fields?)
11/ reveal the main driving mechanisms of your results, provide deeper synthesis and revel some more original/significant findings
12/ propose methods to commercialize the findings obtained, who will benefit from your work?
13/ propose some improvements and direction for future research
Methods:
14/ the method must be presented in such a way that it can be reproduced anytime, by anyone, anywhere (do not create obstacles like referring to specific location etc.)
15/ each material/reactant and apparatus used needs to be presented in detail (serial number, setup, manufacturer, country of origin, purity etc.)
Conclusions:
16/ do not repeat your methods and results again and again, please understand that the Conclusion chapter is not a summary of your work, present only original and industrially significant revelations that have the potential to expand the horizon of human knowledge (higher level of generalization is mandatory)
17/ clearly indicate whether the research hypotheses tends to be confirmed or not
Author Response
We would like to express many thanks to all reviewers for their very useful recommendations to our manuscript. Nearly all of them were implemented into the text and helped to improve the quality of our paper. Our reaction to the reviewers' comments are lower.
Reviewer No. 1
Comments and Suggestions for Authors
Title:
1/ significant shortening mandatory
2/ condensate the main findings obtained from your modelling into a short and groundbreaking claim
Abstract:
3/ better follow the established schema of writing academic Abstract: A/ introduction (urgency and significance of the research hypothesis); B/ principles of the methods used + key results; C/ conclusions (commercial and environmental impacts)
4/ there is no reason to go into detail and present the results obtained under specific reaction conditions, rather provide a synthesis of the results obtained
5/ originality of these "discoveries" is questionable, similar findings can be deduced from existing literature, better highlight the novelty of your discovery (reveal the mechanisms behind your results)
Introduction:
6/ make sure that this chapter fully introduces any reader into to the topic, explain all the botanical terms, units, Latin and Greek letters, abbreviations and the whole context that is necessary for anyone (including experts from other disciplines) to understand the following chapters
7/ deeper review the latest findings in the field, comment on the industrial and economic point of view
8/ the research hypothesis is not clearly stated, better justify the urgency and importance of its investigation (clearly identify those who will benefit from the findings obtained)
Results:
9/ avoid data overkill, present only the most most industrially and environmentally important results
Discussion:
10/ show more self-criticism to your work (can all the methods and results be fully trusted? what are the weaknesses of the methods used? where do the main measurement inaccuracies arise? what are the limitations from a commercial point of view? are the lessons learned transferable to other fields?)
11/ reveal the main driving mechanisms of your results, provide deeper synthesis and revel some more original/significant findings
12/ propose methods to commercialize the findings obtained, who will benefit from your work?
13/ propose some improvements and direction for future research
Methods:
14/ the method must be presented in such a way that it can be reproduced anytime, by anyone, anywhere (do not create obstacles like referring to specific location etc.)
15/ each material/reactant and apparatus used needs to be presented in detail (serial number, setup, manufacturer, country of origin, purity etc.)
Conclusions:
16/ do not repeat your methods and results again and again, please understand that the Conclusion chapter is not a summary of your work, present only original and industrially significant revelations that have the potential to expand the horizon of human knowledge (higher level of generalization is mandatory)
17/ clearly indicate whether the research hypotheses tends to be confirmed or not
The comments by the Reviewer No. 1 are mostly related to the general principles and correct way of academic-style writing. The comments were very useful to reconsider the specific sections of the paper and improve them.
We hope that most of the recommendations were well understood and we implemented them in the optimal way.
Besides other changes we shortened the title, stated and proved our hypothesis, reduced the amount of the results (avoiding overkilling the data), modified introduction and discussion, and rewrote the conclusion. We think that we do not need to answer each point mentioned by the reviewer.

Reviewer 2 Report
Dear authors,
please find my comments in the attached file.
Regards

Author Response
We would like to express many thanks to all reviewers for their very useful recommendations to our manuscript. Nearly all of them were implemented into the text and helped to improve the quality of our paper. Our reaction to the reviewers' comments are lower.
Reviewer No. 2
Comments by the reviewer were inserted in the text. Hence, we tried to incorporate nearly all of them.
We would like to explain that not all presented bark traits were outputs from statistical models derived from measured data. Actually, surface area of bark, bark mass as well as specific surface mass were calculated as a combination of the previously constructed models and some further mathematical or physical equations, hence no statistics is available. Therefore, the result is not any fitting of point cloud with background statistical characteristics. We tried to clarify this in the text and description of figure and hope it is now understandable.
Some comments by the Reviewer No 2 are very similar to those from the Reviewer No. 3. Especially those related to large wild herbivory browsing on stem bark. We modified the text in Introduction, Results, Discussion and Conclusions sections to say that bark browsing was previously expressed mostly on the base of area peeled from stem. Quantification of bark browsing from the point of biomass was not possible because of missing bark models that we presented here. Thus, our models for specific surface area of bark can be considered as a novel tool for such estimates. This is one (but not the only one) of possible utilizations of our bark models. We hope that in this context, ideas related to bark browsing by large herbivory is acceptable in this paper.

Reviewer 3 Report
L24: …on the two mentioned variables
L26: Please consider “had” instead of “manifested”
L30: ….at tree and stand level…
L93: Consider changing “regressive” to “regression”
L100-102: ...shows the species order with regard to tapering (transition from a cylindrical to a conic shape) in the lower part of the stem as follows: ….
L97: The chapter on Materials and methods is missing its expected position! Instead it has been moved after the Discusion section. Also, more references on the use of allometric equations are strongly advised. What were the reasons behind the choice of the specific models and not others? Did you experiment with other models during the data analysis?
Figure 1: Could you provide a cpmbined 4-in-1 for the four species? Each species could be represented by a different color, so that the readership can identify differences more easily.
Figure 2: Same recommendation as for Figure 1.
L118: … i.e. the ratio between…
Tables: In order to avoid “long” tables and given that all P-values are highly significant, could you provide another table layout, where the P-values will be explained once and omitted from multiple positions?
L160: ..exceptionally higher than that.
L161… calculated the total bark mass, staring from the ground level up to this height.
L165: Consider changing “is nearly” to “is in the “range of”
L173 … were found for….
L174… with a D0 of ….
What are the problems of deer browing and…grazing
L215: measured approximately 2.5mm …. and doubled in trees…
L224: … was the bio-socialogical…
L225: “co-dominant” vs “codominant”
L244: More elaborate literature review is needed on this point. Furthermore, depending on game and the existing fauna qualily and availability the extend of browsing damage can vary siginifacantly.
L294: …that was used…
L301: ..of at least 90%...
L315: Table 5. There is no need for the extra grouping of “Mean value and……of”. Instead, you could consider using “Mean tree height (SD)” and Mean Diameter D0 (SD). Also consider using “Altitude range”. Finally, I am not sure that the included note is needed.
L329: “More sections were taken from higher trees”
L389: The formula was
L410-411: Can you justify it by means of literature?
L443: One full stop is sufficient at the spp (not spp..)
I am really worried about the take-home message that this manuscript will deliver. From a certain perspective it tries to present bark as a forage alternative/source however this can be very tricky. The paper can be published containing only the modelling part but not extending to red deer forage management. Range management scientists have tried to address this point in pastures, but they provide, data on grazing capacity for large (cows) and small units (goats, sheep). My worry is that in their effort to provide some original research work, the authors don’t pay the necessary attention to the adverse effects of deer browsing, neither in quantitative nor in qualitative terms.
Author Response
We would like to express many thanks to all reviewers for their very useful recommendations to our manuscript. Nearly all of them were implemented into the text and helped to improve the quality of our paper. Our reaction to the reviewers' comments are lower.
Reviewer No. 3
The comments related to specific mistakes in the text (listed by the Reviewer No 3.) were accepted and incorporated in the text.
Comments and Suggestions for Authors
L24: …on the two mentioned variables
L26: Please consider “had” instead of “manifested”
L30: ….at tree and stand level…
L93: Consider changing “regressive” to “regression”
L100-102: ...shows the species order with regard to tapering (transition from a cylindrical to a conic shape) in the lower part of the stem as follows: ….
L97: The chapter on Materials and methods is missing its expected position! Instead it has been moved after the Discussion section.
Sorry, but the Plants journal requires exactly this order of the sections.
Also, more references on the use of allometric equations are strongly advised. What were the reasons behind the choice of the specific models and not others? Did you experiment with other models during the data analysis?
Thank you for the comment. We were considering it but since there is abundant literature on allometric models we think it would be just a repetition. We have been doing biomass models (allometric relations) for tree components within the last nearly 20 years, having a lot of papers published. After long-term experience of ours (but also according to many other authors) we used recently just the presented kinds of models – they showed to perform well in data description and are rather simple for potential users.
Figure 1: Could you provide a combined 4-in-1 for the four species? Each species could be represented by a different color, so that the readership can identify differences more easily.
Figure 2: Same recommendation as for Figure 1.
As for merging four diagrams in Fig. 1 and Fig. 2. – we accepted it, getting a nice visualization. Really thanks for the great idea!
L118: … i.e. the ratio between…
Tables: In order to avoid “long” tables and given that all P-values are highly significant, could you provide another table layout, where the P-values will be explained once and omitted from multiple positions?
Thank you for the comment. We partially implemented your suggestions. We deleted significant P values from the tables where all were below 0.001 and we left the P values, where not all were below 0.001.
L160: ..exceptionally higher than that.
L161… calculated the total bark mass, staring from the ground level up to this height.
L165: Consider changing “is nearly” to “is in the “range of”
L173 … were found for….
L174… with a D0 of ….
What are the problems of deer browing and…grazing
L215: measured approximately 2.5mm …. and doubled in trees…
L224: … was the bio-socialogical…
L225: “co-dominant” vs “codominant”
L244: More elaborate literature review is needed on this point. Furthermore, depending on game and the existing fauna quality and availability the extend of browsing damage can vary siginificantly.
We modified the text to explain issues about bark browsing. We conducted research on bark browsing previously (e.g. Konôpka, B.; Pajtík, J.; Shipley, L.A. Intensity of red deer browsing on young rowans differs between freshly-felled and standing individuals. For. Ecol. Manag. 2018, 429, 511–519) and we are sure that the new models for specific surface mass are very useful for quantification of browsed bark.
L294: …that was used…
L301: …of at least 90%...
L315: Table 5. There is no need for the extra grouping of “Mean value and……of”. Instead, you could consider using “Mean tree height (SD)” and Mean Diameter D0 (SD). Also consider using “Altitude range”. Finally, I am not sure that the included note is needed.
Thank you for the comment. We changed the tables based on your suggestions. The explanatory note was erased according to the recommendation.
L329: “More sections were taken from higher trees”
L389: The formula was
L410-411: Can you justify it by means of literature?
L443: One full stop is sufficient at the spp (not spp..)
I am really worried about the take-home message that this manuscript will deliver. From a certain perspective it tries to present bark as a forage alternative/source however this can be very tricky. The paper can be published containing only the modelling part but not extending to red deer forage management. Range management scientists have tried to address this point in pastures, but they provide, data on grazing capacity for large (cows) and small units (goats, sheep). My worry is that in their effort to provide some original research work, the authors don’t pay the necessary attention to the adverse effects of deer browsing, neither in quantitative nor in qualitative terms.
As for the take-home message, we added more explanation in the Conclusions section.
Considering issues on wild large herbivory browsing on stem bark:
We modified text in the Introduction, Results, Discussion and Conclusions sections to say that previously bark browsing was expressed mostly on the base of the peeled area on stem. And quantification of bark browsing impact on biomass base was not possible because of missing bark models. Our models on specific surface area of bark would be a novel tool for such estimates. This is one (but not the only one) of possible utilizations of our bark models. We hope that in this context, ideas related to bark browsing by large herbivory are acceptable in this paper.

Round 2
Reviewer 2 Report
Dear Authors,
Thank you for taking my comments into account.
Regards
Author Response
Thanks,
we have no other comments to Reviewer 2.
Reviewer 3 Report
Congratulations! You have made very good work in incorporating the recommendations and modifying the text. Two easy points to consider:
- The methodology section must be moved before the "Results".
- I don't doubt neither your experience on allometric equations nor the very extensive literature on the topic. However, one or two references (possibly handbooks) might prove very beneficial to younger researchers who, during reading your article, might be interested in these "weird" equations.
Once again congratulations for your work, efforts and flexibility in addressing the provided comments.
Author Response
As for the first comment – we could not fulfil it, since we must follow the instructions of the Plants journal.
To the comment number two – we added three citations related to the approaches in our modelling and calculations.